# The Effect of Birch Pollen Immunotherapy on Apple and rMal d 1 Challenges in Adults with Apple Allergy

**DOI:** 10.3390/nu12020519

**Published:** 2020-02-18

**Authors:** Johanna van der Valk, Birgit Nagl, Roy Gerth van Wljk, Barbara Bohle, Nicolette de Jong

**Affiliations:** 1Department of Internal Medicine, section Allergology & Clinical immunology, Erasmus Medical Centre, 3000CA Rotterdam, the Netherlands; r.gerthvanwijk@erasmusmc.nl (R.G.v.W.); n.w.dejong@erasmusmc.nl (N.d.J.); 2Department of Pulmonary Medicine Franciscus Gasthuis & Vlietland, 3045PM Rotterdam, The Netherlands; 3Department of Pathophysiology and Allergy Research, Medical University of Vienna, A-1090 Wien, Austria; birgit.nagl@meduniwien.ac.at (B.N.); barbara.bohle@meduniwien.ac.at (B.B.)

**Keywords:** allergy, food, immunotherapy, birch pollen, apple, Mal d 1

## Abstract

Background: A proportion of patients allergic to birch pollen are also allergic to pit fruit. The objective of this study was to investigate the effect of immunotherapy with birch pollen on birch-pollen-related apple allergy. Method: Patients with birch pollen immunotherapy underwent a skin-prick test with birch pollen, apple and rMal d 1, global assessments and nasal challenges with birch pollen, open food challenge with apple and a double-blind, placebo-controlled test with rMal d 1 at the start of and during the immunotherapy. Measurements of specific IgE in response to Bet v 1 and rMal d 1 and IgG4 in response to Bet v 1 and rMal d 1 took place. Results: Six of eight patients demonstrated an improvement of nasal challenge test results and all patients improved on global assessment during the immunotherapy. The median oral dose of apple required to elicit a reaction increased but was not statistically significant. The patients showed a decrease in skin-prick test values in response to birch pollen (1.05 to 0.36), apple (0.78 to 0.25) and rMal d 1 (0.51 to 0.10) with *p*-values of 0.04, 0.03 and 0.06, respectively and a decrease of specific IgE in response to Bet v 1 (10.66 kU/L to 5.19 kU/L) and rMal d 1 (0.99 to 0.61 kU/L) with *p*-values of 0.01 and 0.05, respectively. Only the median specific IgG4 value to Bet v 1 increased from 0.05 to 1.85 mg/L (*p*-value of 0.02) and not to IgG4 rMal d 1 (0.07 to 0.08 kU/L). Conclusion: The beneficial effects of immunotherapy for birch pollen were accompanied by a limited effect on apple allergy.

## 1. Introduction 

Allergen-specific immunotherapy (AIT) is currently the only treatment for IgE-mediated birch pollen (BP) allergic patients that results in long-term clinical tolerance. [1] Efficacy of AIT with BP has been demonstrated in many clinical trials [2]. Successful AIT for respiratory allergies is characterized by a decrease in nasal symptoms and lowering of nasal reactivity. Nasal reactivity to allergen can be measured with nasal challenge tests [3,4]. A substantial proportion of patients allergic to BP are also allergic to stone fruits (e.g., apple, pear, peach), which is caused by PR-10 allergens [5]. The major allergen in birch pollen (*Betula verrucosa*) is a pathogenesis-related (PR)-10 protein, Bet v 1; the major allergen in apple (*Malus domesticus*) is Mal d 1, which is also a PR-10 protein. These allergens are responsible for cross-reactivity between birch pollen and apple and appear to be the most prevalent cause of oral allergy in BP-allergic patients. BP-associated apple allergy is mainly characterized by mild local symptoms, called oral allergy syndrome (OAS). rMal d 1 (PR-10 protein) is responsible for cross-sensitivity between birch pollen (Bet v 1) and apple. The objective of this study was to investigate the effect of BP-AIT on apple allergy, measured with open food challenges (OFCs) with apple and double-blind, placebo-controlled challenge (DBPCC) with rMal d 1. The secondary aim is to investigate the effect of BP-AIT on the sIgE and sIgG4 levels.

## 2. Material and Methods

### 2.1. Subject Selection and Study Design

Patients (>18 years) visiting the outpatient clinic of Allergology of the Erasmus Medical Centre Rotterdam with BP allergy and starting with birch pollen immunotherapy (BP-AIT) in 2012 and 2013 were asked to participate in this study, after medical ethical approval (obtained on 26 July 2012 (NL40576.078.12).

The first study visit (before the start of the immunotherapy) consisted of filling out a dietary history questionnaire on apple consumption and allergic symptoms. A skin-prick test (SPT) with BP, apple and rMal d 1 and a nasal challenge with BP were done. Thereafter, a DBPCFC was performed with rMal d 1 and an OFC with apple. The challenges and SPTs were repeated after 12 and 24 months. The effect of immunotherapy was measured comparing results from a global assessment questionnaire (after 6, 12, 18 and 24 months) and nasal challenges (at the start of the study, after 12 and 24 months). Blood sampling was performed at the start and after 1, 6, 12, 18 and 24 months of BP-AIT. Antihistamines were withdrawn three days before SPT; the nasal and oral challenges and corticosteroids were discontinued 3 weeks before the nasal challenge tests. All subjects gave their informed consent for inclusion before they participated in the study. 

### 2.2. Immunotherapy With Birch Pollen

Patients received a routinely-performed subcutaneous allergen-specific treatment with registered BP allergen extracts for subcutaneous administration (Alutard SQ 197 Bomen-3 Alutard Pollen RVG 16445).

The BP-AIT in the patients started in the period from September to mid-December either in 2012 or 2013. The immunotherapy consisted of a build-up and a maintenance phase. During the build-up phase, the patients received five injections with a 30-min interval, subsequently, seven increasing doses were given at a weekly interval. The maintenance phase consisted of one injection per month during the next 4–5 years. 

### 2.3. Questionnaires

#### 2.3.1. Dietary History Questionnaire

We used dietary history questionnaires specifically designed for this study. The first part of the questionnaire contained questions about apple consumption and possible allergic symptoms after eating apples. The second part of the questionnaire comprised questions about possible allergies to other stone fruits.

#### 2.3.2. Global Assessment Questionnaire

The efficacy of BP-AIT was evaluated by a global assessment questionnaire. This easy to use tool evaluates the patient’s reported effect of BP-AIT based on rating of general improvement on a 6-point ordinal scale (much worse (-2 points), worse (-1 points), unchanged (0 points), better (1 point), much better (2 points), completely recovered (3 points)) [6]. 

### 2.4. Skin-Prick Tests 

The patients underwent an SPT with extracts of BP, apple (Golden Delicious), rMal d 1, a positive control (histamine 10 mg/mL ALK-Abello, Nieuwegein, the Netherlands) in duplicate and a negative control. All the extracts, except BP (ALK 10.000 BU), were made according to a previously described method [7]. The apple SPT was performed using the prick-to-prick method.

SPTs with BP extract and rMal d 1 (20 µL/mL), were performed by applying a drop of the allergen extract on the skin of the volar part of the forearm. The extract was pierced through the skin barrier with a lancet. Twenty minutes after the skin tests, the contours of the wheal were encircled with a fine-tip pen and transferred to a record sheet by translucent tape. The area of the wheals was determined by using a scanner device (Hewlett Packard 2400c) in combination with software previously developed: Precise Automated Area Measurement of Skin Test (PAAMOST). The area of the allergen-induced wheal was divided by the mean area of the histamine-induced wheals. This ratio was defined as the histamine equivalent prick (HEP) index area. An average HEP-index area ≥ 0.4 was considered positive [8].

### 2.5. Measurement of Allergen-Specific Antibodies

Specific IgE (sIgE) in response to Bet v 1 and rMal d 1 and sIgG4 in response to Bet v 1 and rMal d 1 were measured by using an ImmunoCAP (Thermo Fisher Scientific, Nieuwegein, the Netherlands).

### 2.6. Nasal Challenge Tests

To determine the clinical efficacy of BP-AIT, nasal challenges were performed as described previously [9,10,11,12]. In short: four increasing doses of allergen extract (3, 30, 300, 3000 BU/mL) were applied intranasally at 10-min intervals after the control with PBS containing human serum albumin 0.03% and benzalkonium chloride 0.05% (ALK Abelló). The allergen extract was sprayed into each nostril with a nasal pump spray, delivering a fixed dose of 0.125 mL solution. [13] The nasal response was measured 10 minutes after each step of the challenge by a symptom score according to Lebel et al. [14]. This scoring system of the nasal challenge was graded in points and the total score ranged from 0 to 11 points, where 0 stands for absence of symptoms and 11 for a maximum of symptoms.

### 2.7. Oral Challenge Tests

#### 2.7.1. Open Food Challenge With Apple (Golden Delicious)

The OFC with apple was performed following a challenge protocol of Skypala et al. adapted from published challenges involving fruits and vegetables [15]. The OFC microwave apple started with a ‘chew and spit’ phase of a 5 g dose of apple followed by a ‘chew and swallow’ phase with 5, 10 and 15 g doses of apple (doses 1–4). The same steps were repeated with unprocessed apple (doses 5–8), and finally the remainder of the apple was given (dose 9). A possible reaction after consumption of the remainder of the apple, or no reaction, was scored as dose 10. The symptom intensity score was used, ranging from 0–3, 0 for no symptoms, 1 for mild symptoms, 2 for moderate symptoms or 3 for severe symptoms. The challenge was considered positive and discontinued if objective signs of any grade or a subjective symptom graded > 2 developed. 

#### 2.7.2. Double-Blind, Placebo-Controlled Challenge Test With rMal d 1

For the double-blind, placebo-controlled oral challenge (DBPCC) tests, rMald d 1 was used, produced under GMP conditions by Biomay AG, Vienna, Austria. The rMal d 1 was stored in 20 mM sodium carbonate buffer (pH 9.0). The recombinant protein is tasteless and colorless. The challenge test and assessment of symptoms was carried out according to a previous study [16]. First, 250 uL of 0.9% sodium chloride solution was applied to rule out any nonspecific reactivity. Thereafter, 250 uL of increasing doses of rMal d 1 (dose 1, 5 µg; dose 2, 10 µg; dose 3, 20 µg; dose 4, 50 µg; dose 5, >50 µg or no reaction) was administered below the tongue at ten-minute intervals. Placebo challenges were carried out with 250 uL solutions of 0.9% sodium chloride sublingually. The administration of rMal d 1 and placebo were blindly administered randomly with a weekly interval. There were no stop criteria, and all patients completed the DBPCFC test to step 5, unless not medically justified or unethical or if the patient refused to continue the test. All individuals were monitored for 60 minutes after allergen exposure. 

### 2.8. Statistical Analyses

Most of the study results were reported as descriptive because of the low number of patients in this study. The Wilcoxon signed-rank test was used to investigate the difference between median global assessment score, nasal challenge test score, SPT, sIgE and IgG4 tests, and the challenge doses with apple and Mal d 1 results over time. Here, *p*-values of *p* < 0.05 were considered as statistically significant. Statistical analyses were done using SPSS statistics 21.

## 3. Results 

### 3.1. Study Population

In total, 12 patients with an allergy to BP and starting BP-AIT were willing to participate in this study. Of these patients, eight (two males, six females) finished the whole study program and four patients discontinued the study because of reasons such as the burden of blood sampling or the time-consuming challenge procedures. The age of the patients varied between 22 and 56 years (mean 38.4 years). Of the patients, six (75%) reported symptoms after consuming apples. The most frequently reported symptom was ‘the oral allergy syndrome’. Only one patient experienced gastrointestinal and skin symptoms. Four patients had symptoms after apple consumption during the whole year and the others only during the birch pollen season. None of the patients reported symptoms after consumption of processed apple products. Almost all (seven of eight) patients reported symptoms during the consumption of other stone fruits. 

### 3.2. Effect of the Birch Pollen Immunotherapy

The efficacy of BP-AIT was measured with nasal challenge tests and with a global assessment score. Six out of eight patients demonstrated an improvement of the nasal challenge test result during the first two years of the AIT with a minimal improvement of 1 point and a maximal improvement of 10 (median 6) points. The median nasal challenge score improved from 6 points (start of the study) to 3 points (end of the study), *p* = 0.07 (Table 1). The global assessment score showed a decrease of BP-allergic symptoms in all eight patients. The median scores improved after 6 and 24 months of AIT with 1.5 and 2 points (much better), respectively. 

### 3.3. Skin-Prick Tests 

The median SPT HEP-index areas were 1.05 (range 0.31–1.62) in response to BP, 0.78 (range 0–1.64) in response to apple and 0.51 (0–0.61 range) in response to rMal d 1 at the beginning of the study. The median SPT HEP-index areas at the end of the study were all decreased with: 0.36 HEP (range 0–0.75) in response to BP, 0.25 HEP (range 0–0.55) in response to apple and 0.10 HEP (range 0–0.59) in response to rMal d 1. Median SPT HEP results with BP and apple decreased significantly after 2 years of BP-AIT (*p*-values of 0.04 and 0.03, respectively). The trend for the SPT with rMal d 1 did not reach significance (*p* = 0.06), (Figure 1).

### 3.4. Allergen-Specific Antibody Levels

The median sIgE value in response to Bet v 1 was 10.60 kU/L (range 0.66–228 kU/L) at the beginning of the study. Also at the beginning of the study, the median sIgE value in response to rMal d 1 was 0.99 kU/L (range 0.04–94.60 kU/L) The patients showed a significant decrease of the median sIgE in response to Bet v 1 (10.60 to 5.19 kU/L; *p* = 0.01) and a nearly significant decrease of sIgE in response to rMal d 1 (0.99 to 0.61 kU/L; *p* = 0.05) after 2 years of BP-AIT (Figure 2). Only the median sIgG4 value in response to Bet v 1 increased from 0.05 to 1.85 mg/L (*p* = 0.02). There was no significant difference between the median sIgG4 value in response to rMal d 1 between the start and after 2 years of BP-AIT (0.07 to 0.08 kU/L) (Figure 3).

### 3.5. Oral Challenge Tests 

#### 3.5.1. Open Food Challenge With Apple (Golden Delicious)

At the start of the study, five patients showed a positive reaction to an open apple challenge (dose reaction 1–2) and three patients did not react. Of these five patients with a positive challenge at the start of the study, two had a negative challenge with apple after 24 months of AIT with BP, one patient (number 4) reacted on the same dose over time and two patients had a higher-eliciting-dose reaction over time in the open apple challenge (patient 3 from dose 2 to 9; patient 8 from dose 1 to 6). Patient number 7 reacted to apple at the start and after 12 months but these symptoms diminished after 24 months. One patient (patient 6) who had a negative challenge at the start of the study reacted positive after 12 and 24 months on dose 8. The median score at the start of the study was dose 2 and at the end of the study was dose 9.5 (*p* = 0.08) (Table 2 and Figure 4).

#### 3.5.2. Double-Blind, Placebo-Controlled Challenge Test With rMal d 1

At the start of the challenges, four patients out of eight reacted positively to the DBPCC with rMal d 1 (dose reactions: 2–4), and in four patients the challenge was negative. After 24 months, three of these four positive-challenged patients had a negative challenge and one was still positive. Of those who did not react to rMal d 1, one patient (patient 4) became positive after 12 months, however, after 24 months this patient reacted to the verum as well as placebo. So, we labelled the challenge as undecided (UD). The median reaction dose at the start of the study was 4.5, and at the end of the study the median reaction was 5 (*p* = 0.58) (Table 3 and Figure 5).

## 4. Discussion

In all eight patients, 2 years of BP-AIT resulted in an amelioration of nasal symptoms based on the global assessment scores, and nasal reactivity to BP decreased in six of eight patients. Previous studies demonstrated that BP-AIT is a highly effective treatment in individuals with IgE-mediated diseases [17,18,19]. This positive effect of the AIT on the BP allergy in our study was furthermore supported by the decrease of SPT wheal size and sIgE in response to BP and an increase of sIgG4 in response to Bet v 1.

Six patients in this study reported a BP-associated apple allergy at the start of the study and two patients had never experienced symptoms during the consumption of apple, as can be seen in the results of the OFC with apple in Figure 4.

The effect of BP-AIT on apple allergy was investigated with an OFC with apple and a DBPCC with rMal d 1. The DBPCFC is the gold standard for food challenge tests; however, this test is not available for apple [20]. This is the first prospective study that measured the effect of BP-AIT on apple allergy with OFCs over a long time (at 12 and 24 months). Most other studies tested the effect after 12 months [21]. A longer treatment period might be beneficial for the development of clinical tolerance to BP-associated foods. We used microwaved apples for the first doses of the open apple challenges, to prevent possible severe allergic reactions. PR-10 allergens are denatured during microwave treatment. [16] Nevertheless, the majority of the patients (*n* = 5) already reacted to low doses of these pretreated apples. After 24 months of BP-AIT, four out of five could consume small parts of unprocessed apple without symptoms, which might suggest a slight tolerance-inducing effect. However, the results were variable and the increase in median threshold value was not statistically significant.

A previous study by Bohle et al. demonstrated that sublingual challenge tests with rMal d 1 could be applied to evaluate the therapeutic efficacy of BP-AIT on BP-related food allergy [22]. In this study, we demonstrated that, in three patients, the BP-AIT had a positive effect on apple allergy measured with the OFC apple challenge and/or DBPCC with rMal d 1. This is in line with previously performed studies demonstrating that AIT with BP may have an effect on their BP-associated apple allergy in a selected group of patients [23,24].

The standardized dose of recombinant Mal d 1 used in SPTs has the advantage that the concentration of allergen Mal d 1 is comparable at all test moments, in contrast to the natural variation of prick-to-prick method. We have shown in this study a decrease in sensitization (SPT and sIgE) to apple and rMal d 1 during BP-AIT. A study from Asero et al. also showed that 49 patients treated with sublingual BP-AIT had a reduced skin sensitization to apple after 24 months of SIT, in most cases without decrease of oral allergy symptoms during apple challenge test [25]. Apparently, a decrease in sensitization levels to apple allergens during either sublingual or subcutaneous BP-AIT might not always be accompanied by an improvement in allergy. 

Comparing results of challenges with open apple and rMal d 1 before and after SIT showed that half were both positive or negative. However, in the other half, discrepancies between both challenges were seen, most likely due to due to differences in allergens. Yet, in 1999, Son et al. [26] not only found a variation in allergenicity of apple stains, but also found 12 different Mal d 1 clones from seven apple varieties. Moreover, allergens other than Mal d 1 may be relevant in inducing allergic symptoms in response to apple but may not be influenced by BP-AIT.

In contrast to sIgG4 in response to rMal d 1, only the median sIgG4 levels in response to Bet v 1 increased. However, the three patients who showed a positive effect of the BP-AIT on apple allergy showed an increase in sIgG4 in response to r Mal d 1 in our study. Several studies demonstrated that increased amount of sIgG4 during AIT is correlated with the clinical effect of AIT [27,28]. This may explain the lack of clinical effect of BP-AIT on apple allergy. Bohle et al. demonstrated that sublingual immunotherapy with BP did not efficiently alter the immune response to pollen-related food allergens with either rMal d 1-specific IgE and sIgG4 levels or rMal d 1-induced T-cell proliferation change, which may explain why pollen-associated food allergy is frequently not ameliorated by pollen immunotherapy even if respiratory symptoms significantly improve [21]. Again, a possible explanation might be that the apple allergy was based on profilin cross-reactivity between Bet v 2 and rMal d 2 or other cross-reactive allergens [21].

The strength of the study is that we performed OFC with microwaved apple, raw apple and double-blind, placebo-controlled challenges with rMal d 1 at three different time moments of BP-AIT. Together with measuring sIgE, sIgG4 and SPT results, this gives a unique total picture of the effects of BP-AIT on apple allergy.

A limitation of this study is the small study population and the lack of a control group with BP-associated apple allergy without BP-AIT to observe the spontaneous variation of apple allergic symptoms. From that perspective, the study can be seen as a pilot and should be performed in a larger population.

## 5. Conclusions

The beneficial effects of BP-AIT on allergic rhinitis were accompanied by a limited effect on apple allergy in our study, measured by OFC with apple and a DBPCC with rMal d 1. The rise in sIgG4 in response to Bet v 1 was not mirrored by an increase in sIgG4 in response to rMal d 1, which may explain the lack of effect of BP-AIT on apple allergy in some patients. However, the patients showed a median decrease of the SPT-sensitization to BP, apple and rMal d 1 and a decrease of sIgE in response to Bet v 1 and rMal d 1. 

## Figures and Tables

**Figure 1 nutrients-12-00519-f001:**
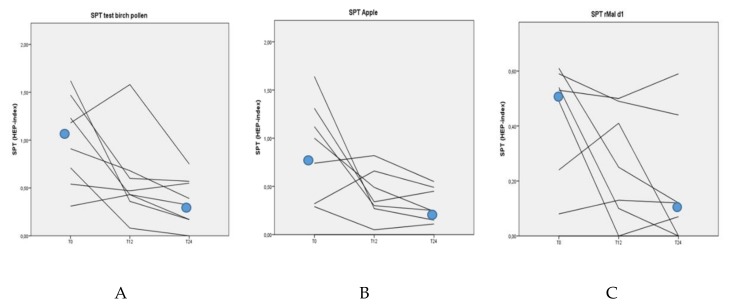
Skin prick test (SPT) results with birch pollen (**A**), apple (**B**) and rMal d 1 (**C**) over time (in months) during birch pollen allergen-specific immunotherapy (BP-AIT).

**Figure 2 nutrients-12-00519-f002:**
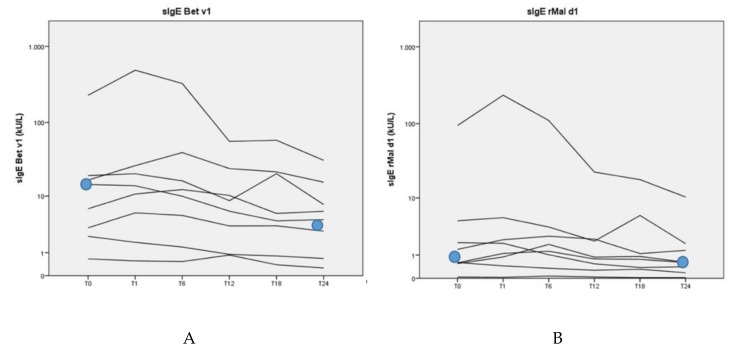
sIgE in response to Bet v 1 (**A**) and rMal d 1 (**B**) over time (in months) during BP-AIT in logarithmic scale.

**Figure 3 nutrients-12-00519-f003:**
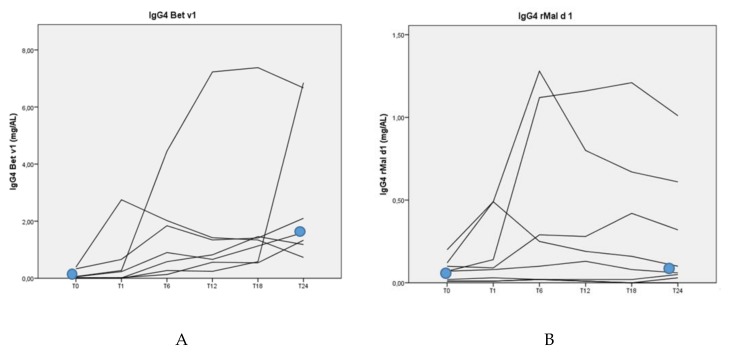
IgG4 in response to Bet v 1 (**A**) and rMal d 1 (**B**) over time (in months) during BP-AIT.

**Figure 4 nutrients-12-00519-f004:**
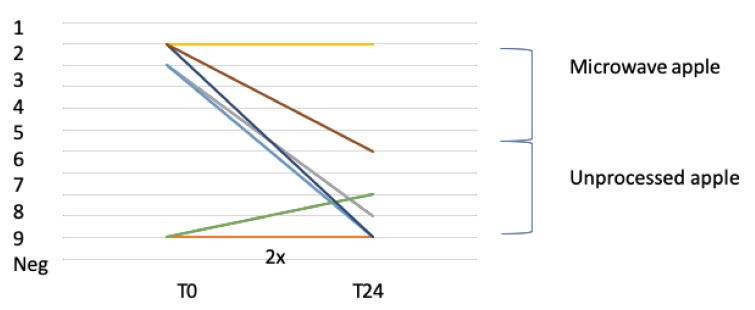
Challenge results of eliciting-dose reaction to open apple before and after 24 months of BP-AIT. Neg. (2×): In two patients the OFCs with unprocessed apple were negative at T0 and T24.

**Figure 5 nutrients-12-00519-f005:**
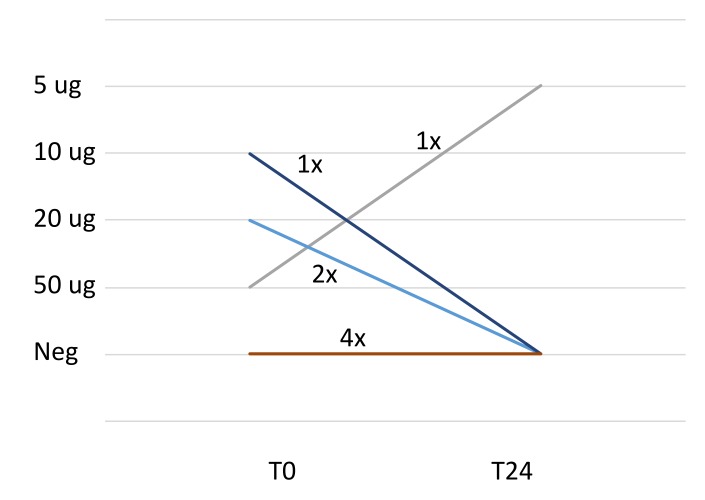
Challenge results of the eliciting-dose from DBPCC with rMal d 1 before and after 24 months of BP-AIT. Neg (4×): In four patients the DBPCC with rMal d 1 was negative at T0 and T24.

**Table 1 nutrients-12-00519-t001:** Effect of the birch pollen immunotherapy measured with nasal challenges.

	Nasal Challenge Score on the Highest Concentration BP Allergen (3000 BU/mL)			
Months AIT	0	12	24	
Patient	Score	Score	Score	Change
1	5	1	4	−1
2	8	5	2	−6
3	6	9	8	2
4	11	2	4	−7
5	6	2	0	−6
6	3	4	1	−2
7	11	6	1	−10
8	4	6	5	+1
Median				
	*6*		*3*	*−3* *(p = 0.07)*

This scoring system of the nasal challenge was graded in points, and the total score ranged from 0 to 11 points, where 0 stands for absence of symptoms and 11 for a maximum of symptoms. The various items scored were sneezing, rhinorrhoea, difficulty in breathing, pruritus of nose and conjunctivitis.

**Table 2 nutrients-12-00519-t002:** Challenge results of eliciting-dose reaction to open apple challenge over time.

	Open Food Challenge (OFC) with Apple					
	Negative or Positive with Eliciting-Dose Reaction					
Months	Before Start AIT	Dose Reaction	After 12 Months AIT	Dose Reaction	After 24 Months AIT	Dose Reaction
Patient						
1	Neg		Pos	2	Neg	
2	Neg		Neg		Neg	
3	Pos	2	Pos	1	Pos	9
4	Pos	1	Pos	1	Pos	1
5	Pos	2	Neg		Neg	
6	Neg		Pos	8	Pos	8
7	Pos	1	Pos	9	Neg	
8	Pos	1	Pos	9	Pos	6

Pos = positive OFC with apple, Neg = negative OFC with apple (no reaction on all doses). Dose reaction for microwave apple: dose 1, ‘chew and spit’ phase of 5 g apple, doses 2, 3 and 4 consisted of a ‘chew and swallow’ phase with doses of 5, 10 and 15 g apple, respectively. Dose reactions of unprocessed apple: dose 5, ‘chew and spit’ phase of 5 g apple; doses 6, 7 and 8 consisted of a ‘chew and swallow’ phase with doses of 5, 10 and 15 g apple, respectively. Dose 9 = remainder of the apple.

**Table 3 nutrients-12-00519-t003:** Challenge results of the eliciting-dose from double-blind, placebo-controlled challenge (DBPCC) with rMal d 1 over time.

	DBPCC with rMal d 1					
	Negative or Positive with Eliciting-Dose Reaction					
Months	Before Start AIT	Dose Reaction	After 12 Months AIT	Dose Reaction	After 24 Months AIT	Dose Reaction
Patient						
1	Pos	20 µg	Neg		Neg	
2	Neg		Neg		Neg	
3	Pos	50 µg	Pos	5 µg	Pos	5 µg
4	Neg		Pos	10 µg	UD	
5	Pos	20 µg	Neg		Neg	
6	Neg		Neg		Neg	
7	Pos	10 µg	Neg		Neg	
8	Neg		Pos	5 µg	Neg	

DBPCC = double-blind, placebo-controlled oral challenge, AIT = allergen immunotherapy, Pos = a positive DBPCC with rMal d 1, Neg = a negative DBPCC with rMal d 1, UD = undecided. Doses of rMal d 1: dose 1 = 0 µg, dose 2 = 5 µg, dose 3 = 10 µg, dose 4 = 20 µg and dose 5 = 50 µg.

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
