# Peer review of "The Effect of Birch Pollen Immunotherapy on Apple and rMal d 1 Challenges in Adults with Apple Allergy"

_nutrients, 2020, doi:10.3390/nu12020519_

Round 1
Reviewer 1 Report
Thank you for the opportunity to review the manuscript titled, The effect of birch pollen immunotherapy on apple and rMal d 1 challenges in apple allergic patients. In this manuscript, the authors report on the results of a double-blind, placebo-controlled test with rMal d 1 at baseline and during immunotherapy, in a population of 8 patients >18 years old. Three patients demonstrated tolerance. Decreases in skin prick testing (SPT) to birch pollen and apple were noted, whereas serum sIgE to these allergens and some of the corresponding components yielded mixed results. Although this is an interesting study, it requires some attention to detail as outlined below.
The title should contain a descriptor about the patient population, such as “adults with apple allergy.” Increasingly, it is unacceptable to use language to define a patient by their disease. Rather than “apple allergic patients”, consider “adults with apple allergy.” The authors report on a study of 8 patients. This is arguably quite a small sample size. As such, the authors may wish to frame it as a pilot study. Alternatively, they may wish to hold off on publication until such a time that more patients are recruited to the study. A reference is needed after the first sentence of the introduction (Lines 31-32). The authors are encouraged to revise “pit fruit” to “pitted fruit.” They may also wish to provide examples of what they mean by pitted- and stone fruits (Lines 35-36). The study was performed in the autumn. Can the authors speculate what might happen if the study was performed in the spring (i.e. birch pollen season)? Amongst the females (n=6), what was the age range? Were some women menopausal or post-menopausal? Using menopausal hormone therapy? If so, did IgE levels and responses differ between these women and the younger women? Does microwaving apple denature the protein? The discussion seems rather “thin” and would benefit from greater integration of the previous literature, discussions about some of the comments above, and clinical implications of the present study. I miss a paragraph on the strengths of the study. The manuscript contains some grammatical, language and syntax errors. Whereas these errors do not impede comprehension, they are nonetheless distracting and should be corrected. Many of the references contain formatting errors.Author Response
Pleas see attachment

Reviewer 2 Report
--GENERAL OVERVIEW--
The submitted manuscript ‘Nutrients-706635’ presents the results of a clinical study on the effect of birch pollen immunotherapy on apple allergy, which is known to have cross-reactive allergens. Both clinical and biochemical results are presented.
Birch pollen allergy improved at both the clinical and antibody levels.
With regard to apple allergy, skin prick test and specific IgE levels showed improvement. Clinical reaction to oral apple challenge improved in several patients (ie. the dose of apple required to elicit a reaction increased). However, 1 to 2 patients appeared to have mildly worsened challenge reaction, which is likely simply be due to natural variability over time. The authors acknowledged the low sample size. The major comment I suggested below is that the description of those oral challenge results, in the Abstract and Discussion, should acknowledge the variability and/or focus on the median. Consistent with this, the authors appropriately concluded at the end of the Abstract that there was a limited effect on apple allergy. The Discussion appropriately contextualised relative to other similar findings.
What is particularly novel is that specific IgG4 to birch pollen increased, yet specific IgG4 to apple allergen did not increase. This helped to explore possible mechanisms for the differences in effectiveness of birch pollen immunotherapy on birch allergy versus allergy to cross-reactive apple allergens.
--SPECIFIC COMMENTS--
ABSTRACT:
Line 18: If there’s room, you could specify ‘birch’ “Bet v 1” and ‘apple’ “rMal d 1”.
Line 21: Rather than, “Three patients demonstrated tolerance to apple”, I think it would be more accurate to write something like: ‘The median oral dose of apple required to elicit a reaction increased, but was not statistically significant’. You can re however you see fit here to conclude on all the oral challenge findings.
RESULTS:
Line 139: Please expand on exactly what is meant by “reported once”.
Line 209: The Results text should also mention patient 8 reacted to apple at 12 but not 24 months.
Table 2:
Using full sentences could make the text in the legend below the table easier to read.
Also, in the y-axis label, should “Neg” be ‘10 (Neg)’?
Figure 3:
On the figure, p=0.11 should be moved underneath the second panel.
Figures 4 and 5:
The figure legend should describe that #X represents the number of patients on the same line.
DISCUSSION:
Line 236: It may be worth mentioning if the patients who showed improvement in the apple open challenge were the same patients who saw improvement in the double-blind placebo controlled challenge with purified apple allergen.
Line 237: Repeating the point I made above for line 21: To more wholly capture the variability, a sentence could be added such as, ‘However, results were variable, and the increase in median value was not statistically significant’.
Line 243: You nicely compared your results to other relevant studies. In the sentence where you mention Asero et al, can you please state how long after immunotherapy they observed a reduction in skin prick reaction.
Thank you for the opportunity to review the manuscript.
Round 2
Reviewer 1 Report
Thanks to the authors for addressing my comments. I have no additional comments at this time.
Reviewer 2 Report
Thank you for revising to address my comments.